# Learning Adaptive Value of Information for Structured Prediction

**David Weiss**
University of Pennsylvania
Philadelphia, PA
djweiss@cis.upenn.edu

**Ben Taskar**
University of Washington
Seattle, WA
taskar@cs.washington.edu

## Abstract

Discriminative methods for learning structured models have enabled wide-spread use of very rich feature representations. However, the computational cost of feature extraction is prohibitive for large-scale or time-sensitive applications, often dominating the cost of inference in the models. Significant efforts have been devoted to sparsity-based model selection to decrease this cost. Such feature selection methods control computation statically and miss the opportunity to fine-tune feature extraction to each input at run-time. We address the key challenge of learning to control fine-grained feature extraction adaptively, exploiting non-homogeneity of the data. We propose an architecture that uses a rich feedback loop between extraction and prediction. The run-time control policy is learned using efficient value-function approximation, which adaptively determines the value of information of features at the level of individual variables for each input. We demonstrate significant speedups over state-of-the-art methods on two challenging datasets. For articulated pose estimation in video, we achieve a more accurate state-of-the-art model that is also faster, with similar results on an OCR task.

## 1 Introduction

Effective models in complex computer vision and natural language problems try to strike a favorable balance between accuracy and speed of prediction. One source of computational cost is inference in the model, which can be addressed with a variety of approximate inference methods. However, in many applications, computing the scores of the constituent parts of the structured model–i.e. *feature computation*–is the primary bottleneck. For example, when tracking articulated objects in video, optical flow is a very informative feature that often requires many seconds of computation time per frame, whereas inference for an entire sequence typically requires only fractions of a second [16]; in natural language parsing, feature computation may take up to 80% of the computation time [7].

In this work, we show that large gains in the speed/accuracy trade-off can be obtained by departing from the traditional method of "one-size-fits-all" model and feature selection, in which a static set of features are computed for all inputs uniformly. Instead, we employ an *adaptive* approach: the parts of the structured model are constructed specifically at test-time for each particular instance, for example, at the level of individual video frames. There are several key distinctions of our approach:

- **No generative model.** One approach is to assume a joint probabilistic model of the input and output variables and a utility function measuring payoffs. The expected value of information measures the increase in expected utility after observing a given variable [12, 8]. Unfortunately, the problem of computing optimal conditional observation plans is computationally intractable even for simple graphical models like Naive Bayes [9]. Moreover, joint models of input and output are typically quite inferior in accuracy to discriminative models of output given input [10, 3, 19, 1].

- **Richly parametrized, conditional value function.** The central component of our method is an approximate value function that utilizes a set of *meta-features* to estimate future changes in value of information given a predictive model and a proposed feature set as input. The critical advantage here is that the meta-features can incorporate valuable properties beyond confidence scores from the predictive model, such as long-range input-dependent cues that convey information about the self-consistency of a proposed output.
- **Non-myopic reinforcement learning.** We frame the control problem in terms of finding a *feature extraction policy* that sequentially adds features to the models until a budget limit is reached, and we show how to learn approximate policies that result in accurate structured models that are dramatically more efficient. Specifically, we learn to weigh the meta-features for the value function using linear function approximation techniques from reinforcement learning, where we utilize a deterministic model that can be approximately solved with a simple and effective sampling scheme.

In summary, we provide a discriminative, practical architecture that solves the value of information problem for structured prediction problems. We first learn a prediction model that is trained to use subsets of features computed sparsely across the structure of the input. These feature combinations factorize over the graph structure, and we allow for sparsely computed features such that different vertices and edges may utilize different features of the input. We then use reinforcement learning to estimate a value function that adaptively computes an approximately optimal set of features given a budget constraint. Because of the particular structure of our problem, we can apply value function estimation in a batch setting using standard least-squares solvers. Finally, we apply our method to two sequential prediction domains: articulated human pose estimation and handwriting recognition. In both domains, we achieve more accurate prediction models that utilize less features than the traditional monolithic approach.

## 2   Related Work

There is a significant amount of prior work on the issue of controlling test-time complexity. However, much of this work has focused on the issue of feature extraction for standard classification problems, e.g. through cascades or ensembles of classifiers that use different subsets of features at different stages of processing. More recently, feature computation cost has been explicitly incorporated specifically into the learning procedure (e.g., [6, 14, 2, 5].) The most related recent work of this type is [20], who define a reward function for multi-class classification with a series of increasingly complex models, or [6], who define a feature acquisition model similar in spirit to ours, but with a different reward function and modeling a variable trade-off rather than a fixed budget. We also note that [4] propose explicitly modeling the *value* of evaluating a classifier, but their approach uses ensembles of pre-trained models (rather than the adaptive model we propose). And while the goals of these works are similar to ours–explicitly controlling feature computation at test time–none of the classifier cascade literature addresses the structured prediction nor the batch setting.

Most work that addresses learning the accuracy/efficiency trade-off in a structured setting applies primarily to inference, not feature extraction. E.g., [23] extend the idea of a classifier cascade to the structured prediction setting, with the objective defined in terms of obtaining accurate inference in models with large state spaces after coarse-to-fine pruning. More similar to this work, [7] incrementally prune the edge space of a parsing model using a meta-features based classifier, reducing the total the number of features that need to be extracted. However, both of these prior efforts rely entirely on the marginal scores of the predictive model in order to make their pruning decisions, and do not allow future feature computations to rectify past mistakes, as in the case of our work.

Most related is the prior work of [22], in which one of an ensemble of structured models is selected on a per-example basis. This idea is essentially a coarse sub-case of the framework presented in this work, without the adaptive predictive model that allows for composite features that vary across the input, without any reinforcement learning to model the future value of taking a decision (which is critical to the success of our method), and without the local inference method proposed in Section 4. In our experiments (Section 5), the "Greedy (Example)" baseline is representative of the limitations of this earlier approach.

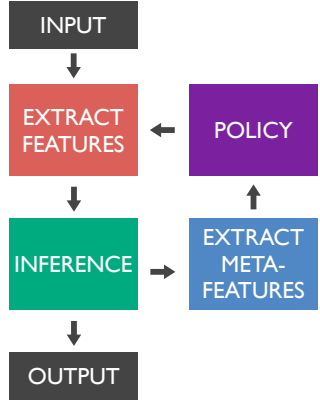

| INPUT |
| EXTRACT FEATURES ← POLICY |
| INFERENCE → EXTRACT META-FEATURES |
| OUTPUT |

**Algorithm 1**: Inference for $\mathbf{x}$ and budget $B$.

**define** an action $a$ as a pair $\langle \alpha \in \mathcal{G}, t \in \{1, \ldots, T\} \rangle$ ;
**initialize** $B' \leftarrow 0, \mathbf{z} \leftarrow \mathbf{0}, \mathbf{y} \leftarrow h(\mathbf{x}, \mathbf{z})$ ;
**initialize** action space (first tier) $\mathcal{A} = \{(\alpha, 1) \mid \alpha \in \mathcal{G}\}$;
**while** $B' < B$ *and* $|\mathcal{A}| > 0$ **do**
    $a \leftarrow \operatorname{argmax}_{a \in \mathcal{A}} \beta^\top \phi(\mathbf{x}, \mathbf{z}, a)$;
    $\mathcal{A} \leftarrow \mathcal{A} \setminus a$;
    **if** $c_a \leq (B - B')$ **then**
        $\mathbf{z} \leftarrow \mathbf{z} + a, B' \leftarrow B' + c_a, \mathbf{y} \leftarrow h(\mathbf{x}, \mathbf{z})$;
        $\mathcal{A} \leftarrow \mathcal{A} \cup (\alpha, t+1)$;
    **end**
**end**

Figure 1: **Overview of our approach.** (Left) A high level summary of the processing pipeline: as in standard structured prediction, features are extracted and inference is run to produce an output. However, information may optionally feedback in the form of extracted meta-features that are used by a control policy to determine another set of features to be extracted. Note that we use stochastic subgradient to learn the inference model $\mathbf{w}$ first and reinforcement learning to learn the control model $\beta$ given $\mathbf{w}$. (Right) Detailed algorithm for factor-wise inference for an example $\mathbf{x}$ given a graph structure $\mathcal{G}$ and budget $B$. The policy repeatedly selects the highest valued action from an action space $\mathcal{A}$ that represents extracting features for each constituent part of the graph structure $\mathcal{G}$.

## 3  Learning Adaptive Value of Information for Structured Prediction

**Setup.** We consider the setting of *structured prediction*, in which our goal is to learn a hypothesis mapping inputs $\mathbf{x} \in \mathcal{X}$ to outputs $\mathbf{y} \in \mathcal{Y}(\mathbf{x})$, where $|\mathbf{x}| = L$ and $\mathbf{y}$ is a $L$-vector of $K$-valued variables, i.e. $\mathcal{Y}(\mathbf{x}) = \mathcal{Y}_1 \times \cdots \times \mathcal{Y}_\ell$ and each $\mathcal{Y}_i = \{1, \ldots, K\}$. We follow the standard max-margin structured learning approach [18] and consider linear predictive models of the form $\mathbf{w}^\top \mathbf{f}(\mathbf{x}, \mathbf{y})$. However, we introduce an additional *explicit* feature extraction state vector $\mathbf{z}$:

$$h(\mathbf{x}, \mathbf{z}) = \operatorname*{argmax}_{\mathbf{y} \in \mathcal{Y}(\mathbf{x})} \mathbf{w}^\top \mathbf{f}(\mathbf{x}, \mathbf{y}, \mathbf{z}). \tag{1}$$

Above, $\mathbf{f}(\mathbf{x}, \mathbf{y}, \mathbf{z})$ is a sparse vector of $D$ features that takes time $\mathbf{c}^\top \mathbf{z}$ to compute for a non-negative cost vector $\mathbf{c}$ and binary indicator vector $\mathbf{z}$ of length $|\mathbf{z}| = F$. Intuitively, $\mathbf{z}$ indicates which of $F$ sets of features are extracted when computing $\mathbf{f}$; $\mathbf{z} = \mathbf{1}$ means every possible feature is extracted, while $\mathbf{z} = \mathbf{0}$ means that only a minimum set of features is extracted.

Note that by incorporating $\mathbf{z}$ into the feature function, the predictor $h$ can learn to use different linear weights for the same underlying feature value by conditioning the feature on the value of $\mathbf{z}$. As we discuss in Section 5, adapting the weights in this way is crucial to building a predictor $h$ that works well for *any* subset of features. We will discuss how to construct such features in more detail in Section 4.

Suppose we have learned such a model $h$. At test time, our goal is to make the most accurate predictions possible for an example under a fixed budget $B$. Specifically, given $h$ and a loss function $\ell : \mathcal{Y} \times \mathcal{Y} \mapsto \mathbb{R}^+$, we wish to find the following:

$$H(\mathbf{x}, B) = \operatorname*{argmin}_{\mathbf{z}} \mathbb{E}_{\mathbf{y}|\mathbf{x}}[\ell(\mathbf{y}, h(\mathbf{x}, \mathbf{z}))] \tag{2}$$

In practice, there are three primary difficulties in optimizing equation (2). First, the distribution $P(Y|X)$ is unknown. Second, there are exponentially many $\mathbf{z}$s to explore. Most important, however, is the fact that we do not have free access to the objective function. Instead, given $\mathbf{x}$, we are optimizing over $\mathbf{z}$ using a *function oracle* since we cannot compute $\mathbf{f}(\mathbf{x}, \mathbf{y}, \mathbf{z})$ without paying $\mathbf{c}^\top \mathbf{z}$, and the total cost of all the calls to the oracle must not exceed $B$. Our approach to solving these problems is outlined in Figure 1; we learn a *control model* (i.e. a policy) by posing the optimization problem as an MDP and using reinforcement learning techniques.

**Adaptive extraction MDP.** We model the budgeted prediction optimization as the following Markov Decision Process. The state of the MDP $s$ is the tuple $(\mathbf{x}, \mathbf{z})$ for an input $\mathbf{x}$ and feature extraction

state $\mathbf{z}$ (for brevity we will simply write $s$). The start state is $s_0 = (\mathbf{x}, \mathbf{0})$, with $\mathbf{x} \sim P(X)$, and $\mathbf{z} = \mathbf{0}$ indicating only a minimal set of features have been extracted. The action space $\mathcal{A}(s)$ is $\{i \mid z_i = 0\} \cup \{0\}$, where $z_i$ is the $i$'the element of $\mathbf{z}$; given a state-action pair $(s, a)$, the next state is deterministically $s' = (\mathbf{x}, \mathbf{z} + \mathbf{e}_a)$, where $\mathbf{e}_a$ is the indicator vector with a 1 in the $a$'th component or the zero vector if $a = 0$. Thus, at each state we can choose to extract one additional set of features, or none at all (at which point the process terminates.) Finally, for fixed $h$, we define the shorthand $\eta(s) = \mathbb{E}_{\mathbf{y}|\mathbf{x}} \ell(\mathbf{y}, h(\mathbf{x}, \mathbf{z}))$ to be the expected error of the predictor $h$ given state $\mathbf{z}$ and input $\mathbf{x}$.

We now define the expected reward to be the adaptive value of information of extracting the $a$'th set of features given the system state and budget $B$:

$$R(s, a, s') = \begin{cases} \eta(s) - \eta(s') & \text{if } \mathbf{c}^\top \mathbf{z}(s') \leq B \\ 0 & \text{otherwise} \end{cases} \qquad (3)$$

Intuitively, (3) says that each time we add additional features to the computation, we gain reward equal to the decrease in error achieved with the new features (or pay a penalty if the error increases.) However, if we ever exceed the budget, then any further decrease does not count; no more reward can be gained. Furthermore, assuming $\mathbf{f}(\mathbf{x}, \mathbf{y}, \mathbf{z})$ can be cached appropriately, it is clear that we pay only the additional computational cost $c_a$ for each action $a$, so the entire cumulative computational burden of reaching some state $s$ is exactly $\mathbf{c}^\top \mathbf{z}$ for the corresponding $\mathbf{z}$ vector.

Given a trajectory of states $s_0, s_1, \ldots, s_T$, computed by some deterministic policy $\pi$, it is clear that the final cumulative reward $R_\pi(s_0)$ is the difference between the starting error rate and the rate of the last state satisfying the budget:

$$R_\pi(s_0) = \eta(s_0) - \eta(s_1) + \eta(s_1) - \cdots = \eta(s_0) - \eta(s_{t^\star}), \qquad (4)$$

where $t^\star$ is the index of the final state within the budget constraint. Therefore, the optimal policy $\pi^\star$ that maximizes expected reward will compute $\mathbf{z}^\star$ minimizing (2) while satisfying the budget constraint.

**Learning an approximate policy with long-range meta-features.** In this work, we focus on a straightforward method for learning an approximate policy: a batch version of least-squares policy iteration [11] based on $Q$-learning [21]. We parametrize the policy using a linear function of *meta-features* $\phi$ computed from the current state $s = (\mathbf{x}, \mathbf{z})$: $\pi_\beta(s) = \operatorname{argmax}_a \beta^\top \phi(\mathbf{x}, \mathbf{z}, a)$. The meta-features (which we abbreviate as simply $\phi(s, a)$ henceforth) need to be rich enough to represent the value of choosing to expand feature $a$ for a given partially-computed example $(\mathbf{x}, \mathbf{z})$. Note that we already have computed $\mathbf{f}(\mathbf{x}, h(\mathbf{x}, \mathbf{z}), \mathbf{z})$, which may be useful in estimating the confidence of the model on a given example. However, we have much more freedom in choosing $\phi(s, a)$ than we had in choosing $\mathbf{f}$; while $\mathbf{f}$ is restricted to ensure that inference is tractable, we have no such restriction for $\phi$. We therefore compute functions of $h(\mathbf{x}, \mathbf{z})$ that take into account large sets of output variables, and since we need only compute them for the particular output $h(\mathbf{x}, \mathbf{z})$, we can do so very efficiently. We describe the specific $\phi$ we use in our experiments in Section 5, typically measuring the self-consistency of the output as a surrogate for the expected accuracy.

**One-step off-policy $Q$-learning with least-squares.** To simplify the notation, we will assume given current state $s$, taking action $a$ deterministically yields state $s'$. Given a policy $\pi$, the value of a policy is recursively defined as the immediate expected reward plus the discounted value of the next state:

$$Q_\pi(s, a) = R(s, a, s') + \gamma Q_\pi(s', \pi(s')). \qquad (5)$$

The goal of $Q$-learning is to learn the $Q$ for the optimal policy $\pi^\star$ with maximal $Q_{\pi^\star}$; however, it is clear that we can increase $Q$ by simply stopping early when $Q_\pi(s, a) < 0$ (the future reward in this case is simply zero.) Therefore, we define the *off-policy* optimized value $Q_\pi^\star$ as follows:

$$Q_\pi^\star(s_t, \pi(s_t)) = R(s_t, \pi(s_t), s_{t+1}) + \gamma \left[ Q_\pi^\star(s_{t+1}, \pi(s_{t+1})) \right]_+. \qquad (6)$$

We propose the following one-step algorithm for learning $Q$ from data. Suppose we have a finite trajectory $s_0, \ldots, s_T$. Because both $\pi$ and the state transitions are deterministic, we can unroll the recursion in (6) and compute $Q_\pi^\star(s_t, \pi(s_t))$ for each sample using simple dynamic programming. For example, if $\gamma = 1$ (there is no discount for future reward), we obtain $Q_\pi^\star(s_i, \pi(s_i)) = \eta(s_i) - \eta(s_{t^\star})$, where $t^\star$ is the optimal stopping time that satisfies the given budget.

We therefore learn parameters $\beta^\star$ for an approximate $Q$ as follows. Given an initial policy $\pi$, we execute $\pi$ for each example $(\mathbf{x}^j, \mathbf{y}^j)$ to obtain trajectories $s_0^j, \ldots, s_T^j$. We then solve the following

least-squares optimization,

$$\beta^\star = \operatorname*{argmin}_{\beta} \lambda ||\beta||^2 + \frac{1}{nT} \sum_{j,t} \left( \beta^\top \phi(s_t^j, \pi(s_t^j)) - Q_\pi^\star(s_t^j, \pi(s_t^j)) \right)^2, \quad (7)$$

using cross validation to determine the regularization parameter $\lambda$.

We perform a simple form of policy iteration as follows. We first initialize $\beta$ by estimating the expected reward function (this can be estimated by randomly sampling pairs $(s, s')$, which are more efficient to compute than $Q$-functions on trajectories). We then compute trajectories under $\pi_\beta$ and use these trajectories to compute $\beta^\star$ that approximates $Q_\pi^\star$. We found that additional iterations of policy iteration did not noticeably change the results.

**Learning for multiple budgets.** One potential drawback of our approach just described is that we must learn a different policy for every desired budget. A more attractive alternative is to learn a single policy that is tuned to a range of possible budgets. One solution is to set $\gamma = 1$ and learn with $B = \infty$, so that the value $Q_\pi^\star$ represents the best improvement possible using some optimal budget $B^\star$; however, at test time, it may be that $B^\star$ is greater than the available budget $B$ and $Q_\pi^\star$ is an over-estimate. By choosing $\gamma < 1$, we can trade-off between valuing reward for short-term gain with smaller budgets $B < B^\star$ and longer-term gain with the unknown optimal budget $B^\star$.

In fact, we can further encourage our learned policy to be useful for smaller budgets by adjusting the reward function. Note that two trajectories that start at $s_0$ and end at $s_{t^\star}$ will have the same reward, yet one trajectory might maintain much lower error rate than the other during the process and therefore be more useful for smaller budgets. We therefore add a shaping component to the expected reward in order to favor the more useful trajectory as follows:

$$R_\alpha(s, a, s') = \eta(s) - \eta(s') - \alpha \left[ \eta(s') - \eta(s) \right]_+. \quad (8)$$

This modification introduces a term that does not cancel when transitioning from one state to the next, *if the next state has higher error than our current state.* Thus, we can only achieve optimal reward $\eta(s_0) - \eta(s_{t^\star})$ when there is a sequence of feature extractions that never increases the error rate[1]; if such a sequence does not exist, then the parameter $\alpha$ controls the trade-off between the importance of reaching $s_{t^\star}$ and minimizing any errors along the way. Note that we can still use the procedure described above to learn $\beta$ when using $R_\alpha$ instead of $R$. We use a development set to tune $\alpha$ as well as $\gamma$ to find the most useful policy when sweeping $B$ across a range of budgets.

**Batch mode inference.** At test time, we are typically given a *test set* of $m$ examples, rather than a single example. In this setting the budget applies to the entire inference process, and it may be useful to spend more of the budget on difficult examples rather than allocate the budget evenly across all examples. In this case, we extend our framework to concatenate the states of all $m$ examples $s = (\mathbf{x}_1, \ldots, \mathbf{x}_m, \mathbf{z}_1, \ldots, \mathbf{z}_m)$. The action consists of choosing an example and then choosing an action within that example's sub-state; our policy searches over the space of *all* actions for *all* examples simultaneously. Because of this, we impose additional constraints on the action space, specifically:

$$z(a, \ldots) = 1 \implies z(a', \ldots) = 1, \quad \forall a' < a. \quad (9)$$

Equation (9) states that there is an inherent *ordering* of feature extractions, such that we cannot compute the $a$'th feature set without first computing feature sets $1, \ldots, a-1$. This greatly simplifies the search space in the batch setting while at the same time preserving enough flexibility to yield significant improvements in efficiency.

**Baselines.** We compare to two baselines: a simply entropy-based approach and a more complex imitation learning scheme (inspired by [7]) in which we learn a classifier to reproduce a target policy given by an oracle. The entropy-based approach simply computes probabilistic marginals and extracts features for whichever portion of the output space has highest entropy in the predicted distribution. For the imitation learning model, we use the same trajectories used to learn $Q_\pi^\star$, but instead we create a classification dataset of positive and negative examples given a budget $B$ by assigning all state/action pairs along a trajectory within the budget as positive examples and all budget violations as negative examples. We tune the budget $B$ using a development set to optimize the overall trade-off when the policy is evaluated with multiple budgets.

| Feature Tier ($T$) | Error (%) | Time (s) | | |
|---|---|---|---|---|
| | | Fixed | Entropy | $Q$-Learn |
| 4 | 44.07 | 16.20s | 16.20s | **8.91s** |
| 3 | 46.17 | 12.00s | 8.10s | **5.51s** |
| 2 | 46.98 | 5.50s | 6.80s | **4.86s** |
| 1 | 51.49 | 2.75s | — | — |
| Best | **43.45** | — | — | **13.45s** |

Table 1: Trade-off between average elbow and wrist error rate and total runtime time achieved by our method on the pose dataset; each row fixes an error rate and determines the amount of time required by each method to achieve the error. Unlike using entropy-based confidence scores, our $Q$-learning approach always improves runtime over *a priori* selection and even yields a faster model that is also more accurate (final row).

## 4 Design of the information-adaptive predictor $h$

**Learning.** We now address the problem of learning $h(\mathbf{x}, \mathbf{z})$ from $n$ labeled data points $\{(\mathbf{x}^j, \mathbf{y}^j)\}_{j=1}^n$. Since we do not necessarily know the test-time budget during training (nor would we want to repeat the training process for every possible budget), we formulate the problem of minimizing the *expected* training loss according to a uniform distribution over budgets:

$$\mathbf{w}^\star = \operatorname*{argmin}_{\mathbf{w}} \lambda ||\mathbf{w}||^2 + \frac{1}{n} \sum_{j=1}^n \mathbb{E}_{\mathbf{z}}[\ell(\mathbf{y}^j, h(\mathbf{x}^j, \mathbf{z}))]. \tag{10}$$

Note that if $\ell$ is convex, then (10) is a weighted sum of convex functions and is also convex. Our choice of distribution for $\mathbf{z}$ will determine how the predictor $h$ is calibrated. In our experiments, we sampled $\mathbf{z}$'s uniformly at random. To learn $\mathbf{w}$, we use Pegasos-style [17] stochastic sub-gradient descent; we approximate the expectation in (10) by resampling $\mathbf{z}$ every time we pick up a new example $(\mathbf{x}^j, \mathbf{y}^j)$. We set $\lambda$ and a stopping-time criterion through cross-validation onto a development set.

**Feature design.** We now turn to the question of designing $\mathbf{f}(\mathbf{x}, \mathbf{y}, \mathbf{z})$. In the standard pair-wise graphical model setting (before considering $\mathbf{z}$), we decompose a feature function $\mathbf{f}(\mathbf{x}, \mathbf{y})$ into unary and pairwise features. We consider several different schemes of incorporating $\mathbf{z}$ of varying complexity. The simplest scheme is to use several different feature functions $\mathbf{f}^1, \ldots, \mathbf{f}^F$. Then $|\mathbf{z}| = F$, and $\mathbf{z}_a = 1$ indicates that $\mathbf{f}^a$ is computed. Thus, we have the following expression, where we use $z(a)$ to indicate the $a$'th element of $z$:

$$\mathbf{f}(\mathbf{x}, \mathbf{y}, \mathbf{z}) = \sum_{a=1}^F z(a) \left[ \sum_{i \in \mathcal{V}} \mathbf{f}_u^a(\mathbf{x}, y_i) + \sum_{(i,j) \in \mathcal{E}} \mathbf{f}_e^a(\mathbf{x}, y_i, y_j) \right] \tag{11}$$

Note that in practice we can choose each $\mathbf{f}^a$ to be a sparse vector such that $\mathbf{f}^a \cdot \mathbf{f}^{a'} = 0$ for all $a' \neq a$; that is, each feature function $\mathbf{f}^a$ "fills out" a complementary section of the feature vector $\mathbf{f}$.

A much more powerful approach is to create a feature vector as the composite of different extracted features for each vertex and edge in the model. In this setting, we set $\mathbf{z} = [\mathbf{z}_u \ \mathbf{z}_e]$, where $|\mathbf{z}| = (|\mathcal{V}| + |\mathcal{E}|)F$, and we have

$$\mathbf{f}(\mathbf{x}, \mathbf{y}, \mathbf{z}) = \sum_{i \in \mathcal{V}} \sum_{a=1}^F z_u(a, i) \mathbf{f}_u^a(\mathbf{x}, y_i) + \sum_{(i,j) \in \mathcal{E}} \sum_{a=1}^F z_e(a, ij) \mathbf{f}_e^a(\mathbf{x}, y_i, y_j). \tag{12}$$

We refer to this latter feature extraction method a *factor-level* feature extraction, and the former as *example-level*.[2]

**Reducing inference overhead.** Feature computation time is only one component of the computational cost in making predictions; computing the argmax (1) given $\mathbf{f}$ can also be expensive. Note

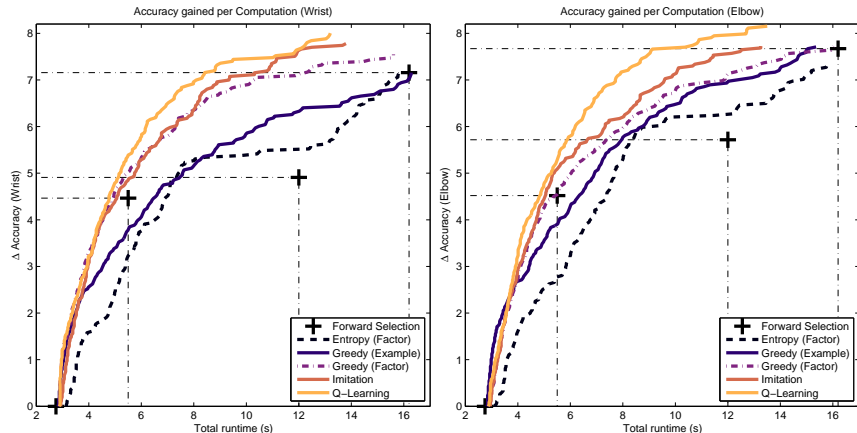

Figure 2: **Trade-off performance on the pose dataset for wrists (left) and elbows (right).** The curve shows the increase in accuracy over the minimal-feature model as a function of total runtime per frame (including all overhead). We compare to two baselines that involve no learning: forward selection and extracting factor-wise features based on the entropy of marginals at each position ("Entropy"). The learned policy results are either greedy ("Greedy" example-level and factor-level) or non-myopic (either our "Q-learning" or the baseline "Imitation"). Note that the example-wise method is far less effective than the factor-wise extraction strategy. Furthermore, $Q$-learning in particular achieves higher accuracy models at a fraction of the computational cost of using all features, and is more effective than imitation learning.

that for reasons of simplicity, we only consider low tree-width models in this work for which (1) can be efficiently solved via a standard max-sum message-passing algorithm. Nonetheless, since $\phi(s, a)$ requires access to $h(\mathbf{x}, \mathbf{z})$ then we must run message-passing every time we compute a new state $s$ in order to compute the next action. Therefore, we run message passing *once* and then perform less expensive local updates using saved messages from the previous iteration. We define an simple algorithm for such *quiescent* inference (given in the Supplemental material); we refer to this inference scheme as $q$-inference. The intuition is that we stop propagating messages once the magnitude of the update to the max-marginal decreases below a certain threshold $q$; we define $q$ in terms of the margin of the current MAP decoding at the given position, since that margin must be surpassed if the MAP decoding will change as a result of inference.

## 5   Experiments

### 5.1   Tracking of human pose in video

**Setup.** For this problem, our goal is to predict the joint locations of human limbs in video clips extracted from Hollywood movies. Our testbed is the MODEC+S model proposed in [22]; the MODEC+S model uses the MODEC model of [15] to generate 32 proposed poses per frame of a video sequence, and then combines the predictions using a linear-chain structured sequential prediction model. There are four types of features used by MODEC+S, the final and most expensive of which is a coarse-to-fine optical flow [13]; we incrementally compute poses and features to minimize the total runtime. For more details on the dataset/features, see [22]. We present cross validation results averaged over 40 80/20 train/test splits of the dataset. We measure localization performance or elbow and wrists in terms of percentage of times the predicted locations fall within 20 pixels of the ground truth.

**Meta-features.** We define the meta-features $\phi(s, a)$ in terms of the targeted position in the sequence $i$ and the current predictions $\mathbf{y}^\star = h(\mathbf{x}, \mathbf{z})$. Specifically, we concatenate the already computed unary and edge features of $y_i^\star$ and its neighbors (conditioned on the value of $\mathbf{z}$ at $i$), the margin of the current MAP decoding at position $i$, and a measure of self-consistency computed on $\mathbf{y}^\star$ as follows. For all sets of $m$ frames overlapping with frame $i$, we extract color histograms for the predicted arm segments and compute the maximum $\chi^2$-distance from the first frame to any other frame; we then also add an indicator feature each of these maximum distances exceeds $0.5$, and repeat for $m = 2, \ldots, 5$. We also add several bias terms for which sets of features have been extracted around position $i$.

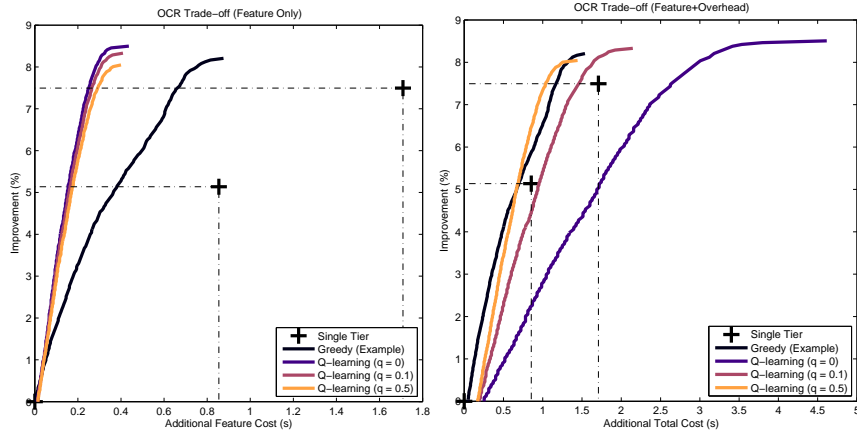

Figure 3: **Controlling overhead on the OCR dataset.** While our approach is is extremely efficient in terms of how many features are extracted (Left), the additional overhead of inference is prohibitively expensive for the OCR task without applying $q$-inference (Right) with a large threshold. Furthermore, although the example-wise strategy is less efficient in terms of features extracted, it is more efficient in terms of overhead.

**Discussion.** We present a short summary of our pose results in Table 1, and compare to various baselines in Figure 2. We found that our $Q$-learning approach is consistently more effective than all baselines; $Q$-learning yields a model that is both more accurate and faster than the baseline model trained with all features. Furthermore, while the feature extraction decisions of the $Q$-learning model are significantly correlated with the error of the starting predictions ($\rho = 0.23$), the entropy-based are not ($\rho = 0.02$), indicating that our learned reward signal is much more informative.

## 5.2 Handwriting recognition

**Setup.** For this problem, we use the OCR dataset from [19], which is pre-divided into 10 folds that we use for cross validation. We use three sets of features: the original pixels (free), and two sets of Histogram-of-Gradient (HoG) features computed on the images for different bin sizes. Unlike the pose setting, the features are very fast to compute compared to inference. Thus, we evaluate the effectiveness of $q$-inference with various thresholds to minimize inference time. For meta-features, we use the same construction as for pose, but instead of inter-frame $\chi^2$-distance we use a binary indicator as to whether or not the specific $m$-gram occurred in the training set. The results are summarized in Figure 3; see caption for details.

**Discussion.** Our method is extremely efficient in terms of the features computed for $h$; however, unlike the pose setting, the overhead of inference is on par with the feature computation. Thus, we obtain a more accurate model with $q = 0.5$ that is $1.5\times$ faster, even though it uses only $1/5$ of the features; if the implementation of inference were improved, we would expect a speedup much closer to $5\times$.

# 6 Conclusion

We have introduced a framework for learning feature extraction policies and predictive models that adaptively select features for extraction in a factor-wise, on-line fashion. On two tasks our approach yields models that both more accurate and far more efficient; our work is a significant step towards eliminating the feature extraction bottleneck in structured prediction. In the future, we intend to extend this approach to apply to loopy model structures where inference is intractable, and more importantly, to allow for features that change the structure of the underlying graph, so that the graph structure can adapt to both the complexity of the input and the test-time computational budget.

**Acknowledgements.** The authors were partially supported by ONR MURI N000141010934, NSF CAREER 1054215, and by STARnet, a Semiconductor Research Corporation program sponsored by MARCO and DARPA.

## Footnotes

[1]While adding features decreases training error on average, even on the training set additional features may lead to increased error for any particular example.

[2]The restriction (9) also allows us to increase the complexity of the feature function $\mathbf{f}$ as follows; when using the $a$'th extraction, we allow the model to re-weight the features from extractions 1 through $a$. In other words, we condition the value of the feature on the current set of features that have been computed; since there are only $F$ sets in the restricted setting (and not $2^F$), this is a feasible option. We simply define $\hat{\mathbf{f}}^a = [0 \ \ldots \ \mathbf{f}^1 \ \ldots \ \mathbf{f}^a \ \ldots \ 0]$, where we add duplicates of features $\mathbf{f}^1$ through $\mathbf{f}^a$ for each feature block $a$. Thus, the model can learn different weights for the same underlying features based on the current level of feature extraction; we found that this was crucial for optimal performance.

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
