[Reviews · NeurIPS 2013]

Submitted by Assigned_Reviewer_2

This is an interesting paper on budgeted structured prediction where the costs of computing features is substantial.

A naive approach to reducing computation would be to add an L1-regularization term to the optimization and only compute the non-zero-weight features. However, the paper is motivated by the observation that for specific instances, some features may not be computationally worthwhile to compute. A controller is constructed using reinforcement learning techniques to sequentially choose the subset of features to evaluate for prediction within the allocated budget.

This is an empirical papers as there is not much in the way of new theory (no theorems, etc.) presented to support the combination of techniques and detailed engineering employed to realize improved efficiency.

A discussion of the literature on budgeted learning for decision trees would be worthwhile since that field of work has similar motivations for discriminative prediction under hard budgets. From that work, the use of information gain rather than pure predictive accuracy might better inform the reward signal employed.

I tend to think that the ideal discriminative model for this setting would have unique weights for each choice of z. This could be re-expressed by expanding the feature space, but much is lost from this level of generalization by employing the proposed much smaller size set of features. There is not much discussion of this, unfortunately.

In (10), why is L2 and not L1 regularization employed?

What is the comparative computational time required to train each of the models?

I have difficulty understanding why the performance is better using a subset of the features. Should I interpret this as a better form of feature selection than e.g., regularization or is there some deeper explanation?

Which objective is the “greedy” baseline using? Would a two-step “greedy” algorithm produce results more similar to the reinforcement learning-based approach?

For the OCR task, how does the performance compare other state-of-the-art approaches for this task?
Summary: This is an interesting paper for instanced-based budgeted prediction that is heavier on engineering a good result than it is on establishing a general theory. It is unclear to me whether the experimental results are significant enough to recommend acceptance (i.e., less than an order-of-magnitude speed improvement).

Submitted by Assigned_Reviewer_5

This paper proposes a framework that adaptively adds features using a Q-learning framework under budget constraints. Their method is able to select the the features at the level of individual variables for each input, and is able to exploit non-homogeneity of the data. They show experiments on two really datasets, where they demonstrate that their algorithm is both more accurate and faster than the baseline algorithms.

1. My main complaint is the clarity of the paper . The proposed procedure is quite complicated, but there is no place in the paper that explicitly describe the overall training and testing procedure, e.g., in an algorithmic box (Figure 1 does a nice job on explaining part of the idea). Adding a clear and compact description of the overall procedure would greatly improve the readability of the paper.

2. I do not understand why the $z$ in Equation (10) when learning the model parameter $w$ is randomly sampled. Should not $z$ be generated from the MDP process?
It seems that the learning of the policy and learning of model parameters is totally independent in the current framework, which is not very desirable. Could you explain this point more?

Quality: median or high.

Clarity: Low or median. See Point 2 above.

Originality: median. Related but different ideas have been proposed before. But this particular framework is novel to my knowledge.

Significance: median or high. The method makes it possible to speedup the structure learning algorithms while selecting useful features under budget constraints. Potentially useful to many practical problems.
Summary: They paper proposes a novel framework for adaptively selecting features under budget constraints, and empirically demonstrate that their algorithm is both more accurate and faster than the baseline algorithms. Their method can potentially helpful for challenging structured prediction problems.

Submitted by Assigned_Reviewer_6

Summary:
This paper proposes an approach to make predictions in a structured output space using a budget constraint on the cost of extracting parts of the feature vector. The authors formulate this as a reinforcement problem and use a simple policy learning algorithm to solve for policy. Experimental results are provided on a challenging human pose estimation and a classical digit classification task and empirically validate the performance of the approach.

Review:
The paper is very well written, the authors do a good job in striking a balance between reviewing previous work and explaining their algorithm. I would have liked to see some of the possible extensions presented at the end of Section 3 being used in the experiments. I could not find any problems with the presented model, the authors provide sufficient details for all parts of the algorithm including test/training time and implementation details.

Experimental results are very well done, the human pose benchmark dataset is challenging, and the authors outperform their baselines while using only a subset of the feature vector. This approach is likely to be of interest to other structured prediction problems as well.

The anonymous submission [2] has to be included as supplementary material. It is discussed in line 102 ff. as "most directly related" and being a "coarse subcase of the framework presented " in [2]. I fear [2] is also under submission at NIPS and both papers are judged individually.

Please state whether source code will be made available or not.

Typos
- line 369 or should be of
- line 371 pterms
- line 202 xit

Summary: Very well written and interesting paper on an important problem in structured output prediction.
Author Feedback

Author rebuttal: We thank the reviewers for their thoughtful and helpful comments and suggestions. We address specific concerns below:

- Information Gain: (R2) Several of the related works we discuss do use information gain as a metric for determining which subsets of features to compute; see e.g. Gao & Koller. But it is not clear how these works would generalize to the setting of structured prediction.

- Unique weights: (R2) In fact, we do use a unique weight for each feature conditioned on the value of z. This is discussed in the footnote on page 6 of the paper (lines 322-323 and 375-377).

- L1 regularization: (R2) In (10), we use L2 regularization because the controller is responsible for choosing which features are used at test-time (so the feature-selection aspect of L1 is redundant.)

- Better performance: (R2) It’s unclear if there is a deeper explanation, but feature selection is one simple explanation for the better performance of the model using fewer features. For each example, there exists an optimal feature set, and test-time per-instance feature selection allows us to better approximate these sets. A similar effect can be seen in stacking architectures that learn to use base classifiers.

- Greedy baseline: (R2) The greedy baseline uses the immediate reward as its objective, without any forward-looking component.

- OCR baselines: (R2) The accuracy of the pixel-only bigram model is 85.7%, which improves to 93.2% when all HoG features are computed; our best result using test-time selection in our bigram model is 94.1% accuracy. This is comparable to state-of-the-art using pixel-only trigram models (e.g. 96.2% in [1]), which are significantly better than pixel-only state-of-the-art bigram models (e.g. 85.0% in [1]).

- Magnitude of effect: (R2) We obtain gains in both speed and accuracy, but the speedups are limited to a certain extent by the overhead of our method (which can be improved in practice.) E.g. on the pose task, our method does yield more than an order of magnitude speedup in terms of features computation time when overhead is not included.

- Clarity: (R5) This is an excellent suggestion that we will include in the final version of the paper.

- Learning the predictor: (R5) The reviewer is correct that the predictor is learned separately from the controller; we first learn the predictor, then learn the control policy based on the outputs from this predictor. One potential avenue of future research is to integrate the two into a single learning process that e.g. would iterate between the two steps.

- Anonymous submission: (R6) The anonymous submission was not submitted to NIPS and relevant portions were included in the supplemental (redundancies have since removed for publication).

[1] Weiss, Sapp & Taskar, Structured Prediction Cascades, http://arxiv.org/pdf/1208.3279v1.pdf.